# Prevalence of Chronic Progressive Lymphedema in the Rhenish German Draught Horse

**DOI:** 10.3390/ani13060999

**Published:** 2023-03-09

**Authors:** Johanna Sievers, Ottmar Distl

**Affiliations:** Institute of Animal Breeding and Genetics, University of Veterinary Medicine Hannover (Foundation), 30559 Hannover, Germany

**Keywords:** chronic progressive lymphedema, feeding, housing, prevalence, Rhenish German draught horse, risk factors

## Abstract

**Simple Summary:**

Chronic progressive lymphedema is a debilitating disease reported in several draught horse breeds. In this study, we analysed the prevalence of this condition and its progression with age in males, geldings, and females in Rhenish German draught horses. In addition, we recorded horse farm-related factors and evaluated their effects on chronic progressive lymphedema. All horses less than 1 year old were not affected by this disease. The first cases were seen in horses that were 1–2 years old. The prevalence of more severe cases increased with an age of 3–6 years. Males were at a 5-fold higher risk for chronic progressive lymphedema than females. Progression with age was faster in males than in females and geldings. Disease progression stopped at a mean age of 16 years in males, 18 years in geldings, and 20 years in females. We found distinct differences among the prevalence of chronic progressive lymphedema among breeding associations. Horses kept on pastures had lower disease scores. Horses with rations in winter without concentrates and rations without hay silage exhibited lower scores for chronic progressive lymphedema. Cannon bone circumference correlated with disease scores, while hoof measures and skinfold diameter at the neck did not. Reducing farm-related risk factors should help to reduce the progression of chronic progressive lymphedema in Rhenish German draught horses.

**Abstract:**

Chronic progressive lymphedema (CPL) is a common disease in Rhenisch German draught horses. The objective of our study was to evaluate the prevalence of this disease and its progression with age in 493 Rhenish German draught horses from different regions in Germany. We employed generalized linear models with binomial, multinomial, and normal distributions to analyse the effects of breeding association, coat colour, sex and age within sex, time of examination, limb on which CPL was recorded, and farm-related factors on disease scores. The occurrence and severity of CPL were significantly influenced by breeding area, sex, coat colour, and interaction of age by sex. Males had significantly higher CPL-scores than geldings (1.92-fold) and females (5.02-fold) as well as a faster (1.85-fold) disease progression per year of life than females (1.44-fold) and geldings (1.25-fold). Regression on age within sex was still significant when age was restricted to a minimum of 7 years in the analysis. Bay horses exhibited 1.77-fold and 2.19-fold higher CPL-scores than chestnut and black horses. Keeping horses on pasture and keeping bedding clean reduced CPL-scores, whereas feeding of hay silage and concentrates during winter increased CPL-scores. Our analysis revealed significant regression coefficients of cannon bone circumference on CPL-scores. Relationships with hoof measures and skinfold diameter at the neck were not found. In conclusion, differences among breeding districts and sexes by age had the largest impact on CPL prevalence and scores. Disease progression was evident up to a mean age of 16 years in males and 20 years in females. The identification of risk factors and their effects on CPL should help to reduce the occurrence and progression of CPL.

## 1. Introduction

Chronic progressive lymphedema (CPL) is a common disease in draught horses [1,2,3,4,5]. The first reports of CPL in draught horses date from the early 1900s and indicate a considerable number of German draught horses were affected [6,7]. Horse breeds mentioned in this first report also include Danish draught, Belgian draught, Shire, and Clydesdale [6]. In a review on the causes of premature leave among 979 coldblooded stallions from the stud in Kreuz, Germany, CPL was found to be the primary cause in 6.6% of the cases [8]. Stallions belonged to different breeds, including Shires (n = 28), Saxon-Thuringians (n = 20), Belgian Draught horses (n = 10), Clydesdales (n = 3), Rhenish Germans (n = 2), and Percherons (n = 1). A survey in the French departments Cluny and Annecy showed a prevalence of CPL in 46–47% of the Ardenn, 10–12% of the Percheron, and 2–3% of the Breton breed [9]. Wallraf et al. investigated the prevalence of CPL in all German draught horse breeds and found the highest prevalence in Rhenish German, Saxon-Thuringian, Schleswig, and Mecklenburg Draught horse breeds, with 96.0, 84.1, 81.4 and 74.5%, respectively [1]. Black Forest and South German showed a lower prevalence, with estimates at 24.0 and 39.0% [1,10]. Studies from Belgium and USA reported CPL in the Belgian draught horse, Shire, Clydesdale, gypsy vanners, as well as the Friesian [2,3,5].

The characteristic clinical signs of CPL include progressive swelling, scaling, hyperkeratosis, crusts, skin folds, nodules, often with exudative wounds and ulcerations, fibrosis of the distal limbs, and, in some cases, verrucous lesions with clearly demarcated calluses and wart-like proliferations [1,2,3,4,5,6,7,8,9,10,11,12]. Secondary bacterial and recurrent parasitic infections may aggravate lymphedema and clinical signs [3,11,12,13]. In contrast to these observations, infestation with *chorioptes* mites was not correlated with the prevalence of the different stages of CPL-lesions in German draught horses [1]. CPL-lesions start to appear at a younger age and prevalence increases with age, regularly leading to limb disfigurement and disability [11]. Pathological examinations of the distal limbs reveal epidermal hyperplasia, dermal oedema, dilated and tortuous lymphatic vessels, development of thick-walled lymphatics, especially in the palmar/plantar region of the fetlock joints, fibrosis of dermis and subcutaneous tissues, and fragmentation and disorganization of elastic fibres [11,14]. The pathological changes in the lymphatic system correspond with the severity of the clinical lesions [11]. Lymphoscintigraphy indicated a delay in lymphatic clearance in the limbs of affected horses [14]. In addition, lymph flow stagnates completely in severely affected limbs [14]. These findings strengthen the hypothesis that the clinical signs of CPL may be due to stagnant lymph flow [14]. Defects in the network of elastic fibres surrounding dermal lymphatics are assumed to be associated with chronic lymphedema [15]. Elastin content of the skin can be quantified by measuring cutaneous desmosine levels. CPL-affected Clydesdale and Shire horses show higher cutaneous desmosine levels than clinically normal horses of the same breeds, but controls of the breed Percheron have significantly higher cutaneous desmosine levels than CPL-unaffected Clydesdale and Shire horses [15]. Mildly CPL-affected horses have the highest cutaneous desmosine levels; however, levels decrease in more severely CPL-affected horses. Accumulation of elastin and amounts of desmosine are highest in superficial dermis in the distal limb and neck [16,17]. However, a study in Belgian draught horse stallions could not confirm the use of cutaneous desmosine levels as a diagnostic aid for CPL [18]. In summary, progressive lymphedema and tissue fibrosis, along with disorganized elastic fibres supporting lymphatic vessels, impair lymphatic clearance and correlate with disease progression; however, the primary causative factors for the development of CPL still remain unknown [13,14,15]. 

CPL is an incurable disease that requires intensive management to slow down the disease progression [13,19]. Treatment of secondary infections is difficult due to severe feathering, treatment failures, and recurrences [19,20]. Clipping of feathering, manual lymph drainage, bandaging with short stretch bandages, and exercise are recommended [19,21]. Removing multiple multifocal verrucous masses using dissection and electrocautery from the distal hindlimb of a draught horse did not result in regrowth after a 24-month period [22].

The occurrence and severity of CPL-lesions showed interactions between sex and age in previous studies [1,10,12,23,24,25]. A study with 37 cases, including mainly German draught horses, three Percherons, one Belgian and one Polish draught horse, and two controls showed a significant correlation between the severity of CPL-lesions and age of the horses [12]. In 912 draught horses, including all different German draught horse breeds, sex and linear regression on age were significant for the prevalence of CPL and number of limbs affected with CPL [1]. Both studies with 431 and 980 records of Belgian draught horses confirmed the significant effects of age and sex by age on CPL severity [2,24]. Stallions tended to develop CPL-lesions quicker than mares [24]. Along with more severe CPL-lesions, cannon bone circumference [12] and skinfold thickness increased [1,10,12,24,25]. Contrasting results were obtained for all German draught horse breeds, where residual correlations among skinfold thickness and CPL prevalence were close to zero in multivariate animal models [1,10,25].

Horse farm-related factors affect the severity of CPL-lesions. Horses kept in outside pens on rubber meadows were less severely affected than those kept on sand or soil [12]. Stable hygiene and stable quality were correlated with the prevalence [10,25] and severity of CPL-lesions [12,25]. A restrictive feeding management with refrain of concentrates and hay silage reduced the risk of developing CPL-lesions [10,26,27].

The large differences in prevalence of CPL among cold blooded breeds and the disposition of draught horses to CPL suggests the involvement of genetics [3,7,8]. Heritability has been estimated for CPL-scores in Belgium draught horses [23] and in South German, Black Forest, Schleswig, Rhenish German and East German (Mecklenburg, Saxon-Thuringian) draught horses for the prevalence of CPL, the different stages of CPL, and the number of affected limbs [1,10,25]. Analyses in Belgian draught horses restricted to horses older than 3 years revealed higher heritability estimates with 0.26, compared to data with horses of all ages with 0.11 [23]. Animal model analyses across German draught horse breeds resulted in heritability estimates of 0.21 for the prevalence of CPL-lesions and 0.24 for the number of limbs affected [10]. Heritability estimates for the South German and Black Forest draught horses were lower than for Schleswig, Rhenish, and East German breeds [25]. Genetic correlations with skinfold thickness (0.28–0.43) and pronounced fetlock tufts of hairs (0.32–0.34) were moderate in German draught horses [10], but inconclusive in Belgian draught horses for both measures, skinfold thickness, and hair diameter due to their large standard errors [23]. A genome-wide scan with 318 microsatellites for 378 German draught horses using multipoint linkage analyses revealed four chromosome-wide significant quantitative trait loci on ECA 1, 9, 16 and 17 [28]. A genome-wide association study using 307,474 single nucleotide polymorphisms (SNPs) for Friesian horses with 26 cases (CPL-affected) and 19 controls did not identify significantly associated loci for CPL [4]. 

The clinical relevance of CPL and its associated impact on animal welfare, health, and reduced life expectancy are factors that contribute to the importance of determining the prevalence of CPL and possible risk factors [5,11]. Previous reports indicated that CPL may have a significant impact on the health of Rhenish German draught horses [7,25] and therefore breeders may be discouraged from keeping this breed. In this way, CPL could be a reason why the population size of the Rhenish German draught horse population continues to decrease below 1000 breeding animals. Therefore, our objective of this study was to examine the current prevalence and severity of CPL as well as the age at onset in Rhenish German draught horses in Germany. In addition, we evaluated risk factors associated with CPL. We employed a CPL scoring system based on Wallraf [25], Affolter [3] and de Keyser et al. [2] to be applied for each limb in order to monitor CPL-scores based on a standardised system. 

At each horse farm visit, we collected data on animal variables and housing, feeding, management, exercise of horses, and hoof care. These data should allow evaluation of associations between CPL and animal as well as horse farm-related variables. Due to the large number of studs across Germany, stud and animal variables were not confounded with each other. Thus, we employed multivariable models to evaluate the effects of risk factors on CPL.

## 2. Materials and Methods

### 2.1. Ethical Approval

The study was performed according to the guidelines of the Declaration of Helsinki and approved by the Institutional Review Board of the University of Veterinary Medicine Hannover (Foundation) and the state veterinary offices from the different German Federal States for North Rhine-Westphalia (registration number 81-02.05.40.19.083), Lower Saxony (registration number 33.8-42502-05-19A465), Thuringia (registration number 22-2684-04-TIH-20-101), Brandenburg (registration number 2347-A-19-1-2020) and Saxony (registration number 25-5131/521/20). The sampling and handling of the horses followed European Union guidelines for animal care and handling and the Guidelines of Good Veterinary Practices.

### 2.2. Sample Collection

The study was performed in cooperation with the Westphalian Breeding Association, the Rhenish Breeding Association, and the North Rhine-Westphalian State Stud. We presented the project to the breeders in several meetings and published articles on the aims and study design in horse breeders’ journals. We invited all Rhenish German Draught horse breeders in North Rhine-Westphalia to participate in the study. In addition, we offered breeders from other German states to be involved in this project. All breeders who agreed to participate in this project received an information letter and were contacted to arrange a visiting appointment for examination of the horses and recording management, housing and feeding data of the horses. All breeders recruited for this study participated voluntarily and gave written informed consent for participating in this research on CPL and possible risk factors. The horse farms were located in Brandenburg, Lower Saxony, Rhineland, Westphalia, and Thuringia.

In total, we examined 493 horses from 96 horse farms (Table 1). Data were collected from February 2019 until October 2021. The mean number of sampled horses per farm was 5.1 ± 8.9 with a minimum of one horse and a maximum of 61 horses per farm. The work of the breeding organizations is limited to a certain region and this means that all horse-farms from a certain region are members of the respective breeding organization. 

The average age of the examined horses was 7.77 ± 6.37 years, with a maximum age of 33.3 years and a minimum of 1–2 months. Figure 1 represents the distribution of the 493 sampled horses by age. The numbers of horses older than 1, 2, 5, 8, 11, and 19 years were 425, 354, 256, 186, 124, and 42, respectively. Most of the horses > 17 years old were females (70.2%) and only a few geldings (15.8%) and males (14.0%). The horses from the oldest group were sampled in Westphalia (n = 23), Thuringia (n = 8), Brandenburg (n = 3), and Rhineland (n = 8). Overall, we sampled 111 males (22.5%), 57 geldings (11.6%), and 325 (65.9%) females. The distribution by sex and age in years is given in Appendix A. Average ages of males, geldings and females were 5.2 ± 5.4 years, 10.2 ± 6.1 years, and 8.2 ± 6.5 years, respectively. The oldest male, gelding, and female reached 21.7, 27.8, and 33.3 years, respectively. 

### 2.3. Data Collection

During each horse farm visit, an interview with the horse owner was conducted. The stud was inspected to record data on the type of stable, the type of use or work applications for the horses, the bedding type, manure management, type of outdoor access and average hours of outdoor exercise per day in autumn and winter or in spring and summer, average hours of pasture grazing, frequency of shoeing, and type of roughage and concentrates fed in the summer and winter months. The answers were documented on a written questionnaire (Appendix A). The questionnaire contained yes-no and closed questions.

### 2.4. Examination of the Horses

Sampling of horses was carried out by a veterinarian. The handheld RFID reader APR600 (Agrident, Barsinghausen, Germany), which can scan transponder types HDX and FDX-B, was used to scan the transponders and record the data of the sampled horse, with a pre-programmed task mode stored on the reader. The scanned transponder number was matched with the equine passport. The Universal Equine Life Number (UELN), name, date of birth, and coat colour of the horse were electronically recorded at the farm. In addition, the skinfold thickness [24] on the neck (Cutimeter, Hauptner, Solingen, Germany) and the hoof conformation were recorded by measuring the length of the dorsal wall, heel length, angle of the dorsal border, and hardness of the hoof horn at the dorsal wall (Shore D, Zwick-Roell, Ulm, Germany) from the left front and right hind limb. Since the differences between left and right hoof are meaningless [25], we measured one front hoof and one rear hoof. To take into account the sides of the body, we chose one hoof from the left and one hoof from the right side. The pigmentation of the hoof colour was documented for the left front hoof and the right hind hoof. 

Classification of CPL signs is shown in Appendix A. The examination of all four limbs included inspection and palpation, starting at the hoofs and ending at the knee or elbow joints. The findings were documented for each limb. Each horse received a final score for each limb and an overall score for all four limbs. 

### 2.5. Statistical Analysis

Statistical analysis of data was performed using SAS, version 9.4 (Statistical Analysis System, Cary, NC, USA, 2022). Descriptive statistics were calculated with the SAS procedures MEANS and distributions and exact binomial 95% confidence intervals (CI) with the SAS procedure FREQ. We used generalised linear mixed models to evaluate associations of CPL-scores with breeding association, sex, month-year-classes at time of examination, coat colour, age, and limb. There was no influence of the investigator to be considered, as all examinations were performed by an experienced veterinarian. Analyses were performed using the SAS procedures GLIMMIX and MIXED. Dependent variates used were the overall CPL-score across all four limbs (CPL-score), the highest CPL-score per horse (CPL-max) on a scale from 0 to 5, the sum of all CPL-scores over all four limbs (CPL-sum), as well as CPL-score > 0, CPL-score > 1, and CPL-score > 2, and CPL-score > 3 as 0/1-variates. Overall CPL-scores were calculated using the CPL-scores per limb. An overall CPL-score of 0 was determined when the sum of CPL-scores per limb was not greater than 1. Overall, a CPL-score of 1 was set when the sum of CPL-scores per limb was >1 and <4 and the maximum CPL-score per limb was 2. Overall, a CPL-score of 2 was deduced when the sum of CPL-scores per limb was >3, at least one limb had a CPL-score of 2, and CPL-scores per limb did not exceed a value of 2. Overall, a CPL-score of 3 was applied when the sum of CPL-scores per limb was >4, at least one limb showed a CPL-score of 3, and no limb had a CPL-score >3. Overall, a CPL-score of 4 was given when the sum of CPL-scores per limb was >7, at least on one limb a CPL-score of 4 was seen, and CPL-scores per limb were <5, and, overall, the CPL-score was 5 when the sum of CPL-scores per limb was >8 and at least one limb showed a CPL-score of 5. 

In the generalised mixed linear models, we employed a binomial distribution function and logit as link function for binary variates, and for categorical variates a multinomial distribution function with a cumulative logit link function or a normal distribution and identity link function to parameterize the probability of an overall or specific CPL-score for individual horses.

The final model 1 included the fixed effects of breeding association, coat colour, sex, and the linear and quadratic regressions on age at examination within sex. The final generalised mixed linear model 1 using cumulative logits for probabilities of CPL-scores (*θ_ijklmn_*) or as quasi-quantititive trait (*y_ijklmn_*) with an identity link function was as follows: *θ_ijklmn_* = cumulative logit *p_ijklmn_* (or *y_ijklmn_*) = *μ*  + breeding association*_i_* + coat colour*_j_* + sex*_k_* + b_1_(age * sex)*_l_*  + b_2_(age^2^ * sex)*_m_* + e*_ijklmn_*
where *μ* is an unknown constant common to all horses in the linear model, and in the multinomial model, unknown constants for the thresholds to which the observed CPL-score belongs, breeding association*_i_* = fixed effect, with i = Brandenburg-Anhalt, Rhenish, Saxon-Thuringian and Westphalian including Lower Saxon; coat colour*_j_* = fixed effect, with j = chestnut, black and bay; sex*_k_* = fixed effect, with k = female, gelding and male; b_1_(age * sex)*_l_*   = linear regression on age in years within the three different sexes; b_2_(age^2^ * sex)*_m_* = quadratic regression on age in years within the three different sexes; e*_ijklmn_* = unknown random residual effect.

CPL-scores and prevalence were not significant between limbs, thus CPL-scores for the single limbs were not used in all following analyses. In addition, we did not find a significant effect for month-year-classes for time at examination. All two-way interactions other than age by sex tested with a stepwise forward and backward selection strategy were not statistically significant and, thus, were omitted in the final model. The stud was tested as a random effect, but did not reach statistical significance. Coat colours distinguished were bay, chestnut, and black. Roan coat colour was not significant as separate fixed effect nor in a two-way interaction with basic coat colours. This final model 1 was employed for all horses sampled and subsets by age groups. Herein, we restricted the analyses for a minimum age of 1, 2, 3, 4, 5, 6, 7, 8, 9, or 10 years in order to test for the importance of the age by sex effects on the different CPL-scores. Analyses of variance were also performed for CPL-max, CPL-sum, CPL-score > 0, CPL-score > 1, and CPL-score > 2, and CPL-score > 3. 

We extended this final model 1 to study horse farm-related factors on CPL-scores. These stud factors were regarded as single additional fixed effects or covariates in model 1. Stud factors included type of stable, outdoor facilities for horses in winter and summer months, bedding type, time interval for cleaning out the stable, type of roughage and concentrate fed in winter months, type of concentrate fed in summer months, other additional feed types for horses, type of hoof care, length of hoof trimming intervals, type of work applications for horses, daily hours of work with horses, and days per week of work with horses. The stud-related factors as additional single factors in model 1 with *p*-values < 0.05 were then tested to reach the final multivariable generalized mixed linear model. Forward selection using the SAS procedure HPQUANTSELECT was performed to test the consistency of the results. In the final model 2, a *p*-value < 0.05 was defined as the significance threshold for inclusion of an effect. 

Animal variables were height at withers, body length, chest circumference, thickness of skinfold at the neck region, and, at the left front and right hind limb, cannon bone circumference, hoof measures including dorsal wall length, heel length, and front angle, and Shore D hardness of hoof horn at the dorsal wall. We tested animal variables as single additional effects using model 1, as well as all animal variables of the body and variables either of the front or of the hind limb with the forward selection procedure HPQUANTSELECT, in order to validate significant associations with CPL-scores. 

## 3. Results

### 3.1. Overall Prevalence by Age and Sex

We found 170 horses without signs of CPL lesions (34.5%) that were classified with a CPL-score of 0 (Table 2). Of the horses with an age of at least 3, 6, and 9 years, 18.4% (CI: 14.1–22.3), 10.2 (CI: 6.7–13.7), and 6.2 (CI: 3.0–9.5) were free from CPL-lesions, respectively. A CPL-score of 3 was most frequently seen (24.3%). Severe signs of CPL with scores 4 and 5 were found in 65 (13.2%) and 6 (1.2%) horses, respectively. In total, 65.5% of all 493 sampled horses had a CPL-score > 0. All horses aged less than one year had a CPL-score of 0. CPL-scores 1–2 became apparent in horses aged >1 and <2 years with one male and three females with a CPL-score of 1 and two males and four females with a CPL-score of 2. The first cases with CPL-scores of 3 were each one gelding aged 1.9 years, one male aged 2.6 years, and one female aged 2.4 years. CPL-scores of 4 became apparent with an age between 6.6 and 8.1 years in four males, between 7.4 and 8.7 years in three geldings, and between 6.3 and 7.2 years in three females. The highest frequencies of CPL-scores 1, 2 and 3–5, were seen in horses aged 1–3 years, 3–6 years, and 12–18 years, respectively. The averages of overall CPL-scores continuously increased with age group from 0 to 2.976. 

Of males, females, and geldings, 43.2%, 36.0, and 8.9, respectively, were free of CPL lesions (Table 3). In horses with at least 3 years of age, 10.3% (CI: 3.1–17.5) of males, 22.6% (CI: 17.5–27.7) of females, and 8.9% (CI: 3.0–19.6) of geldings were free of CPL. Of females 6 and 9 years of age or older, 13.1% (CI: 8.3–17.8) and 6.7% (CI: 2.7–10.7), respectively, had no CPL, whereas all males at this age had CPL lesions. CPL-affected horses of all sexes exhibited scores from 2 to 4 most often. In geldings, a CPL-score of 3 was more frequent than in males and females. The average CPL-score was higher in geldings than in the other sexes because of the highest average age and the missing age group < 1 year.

The proportions of CPL-scores by sex and age in years are shown in Appendix A for males, geldings, and females. At 6 years of age, all males were affected by CPL, while we found a few unaffected mares over 9 years of age and a single unaffected mare at 21.4 years of age. The proportions of horses with CPL-scores >0, >1, >2, and >3 within age classes are given for males, geldings, and females in Appendix A. In males, the steepest increase in the proportion of horses with CPL-scores > 2 and > 3 is observed in the age range 3–5 (age group 3) and 6–8 (age group 4) years. In female horses, the increase in the proportion of horses with CPL-scores > 2 and > 3 is much flatter and extends over a longer age. All males in the age group with 12–17 years reach CPL-scores > 2 and in the age group with 6–8 years already 75% of all males exhibit CPL-scores > 2. In the age group with 6–8 years, 31.8% of all females have CPL-scores > 2, and in the age group with 12–17 years, 54.8% of all females. 

### 3.2. Signalment Data of Enrolled Horses

The most common coat colours were bay (43%), chestnut (33.1%), and bay-roan (12.4%), whereas black (5.5%), chestnut-roan (3.0%), and black-roan (3.0%) were less commonly recorded coat colours (Appendix A). Black horses had, on average, the lowest CPL-score, whereas bay horses had the highest CPL-score.

### 3.3. Generalized Linear Multivariable Model for CPL-Scores

The results of the analyses of the CPL-scores using the final model 1 with a normal and multinomial distribution function showed significant (*p*-value < 0.05) effects for breeding association, sex, coat colour, the linear and quadratic regression on age within sex (Table 4). Final model 1 results for CPL-max, CPL-sum, CPL-score > 0, CPL-score > 1, and CPL-score > 2, and CPL-score > 3 are shown in Appendix A. In addition, we analysed CPL-scores for horses aged ≥1, ≥2, ≥3, ≥4, ≥6, ≥8, ≥10, and ≥12 years using model 1 and a multinomial distribution function (Appendix A). Breeding association was in all age classes significant, whereas sex in horses with 2 years and older was no longer significant. 

Linear and quadratic covariates age within sex were significant even if the minimum age of horses included in the model was greater or equal than 1 to 7 years. The linear covariate age within sex was even significant when the minimum age of horses was restricted to 8 years (Table 5). Linear regression coefficients on age within sex were positive, whereas quadratic regression coefficients were negative for all sexes. The steepest increase in CPL-scores with age was found in males and the smallest in females for all horses, as well as in analyses using restrictions on the minimum age of horses. Quadratic regression coefficients were smaller in males compared to females, whereas geldings had the largest quadratic regression coefficients. Odds ratios (ORs) from the multinomial model with their 95% confidence intervals for differences by one year in age for CPL-scores are given for all horses and horses with a minimum age of 1 to 15 years (Appendix A). The highest OR was found for males with a value of 1.85. Males reached ORs < 1 when the minimum age was ≥11 years and mean age was 16 years, geldings when the minimum age was ≥14 years and mean age was 18 years, and females when the minimum age was ≥16 years and mean age was 20 years. 

Horses of the horse breeding association from Brandenburg-Anhalt showed the lowest average CPL-score, whereas Westphalian horses reached the highest average CPL-score (Table 6). The ORs of Westphalian horses for CPL-scores was 8.16 and 6.05 times higher compared to Brandenburg-Anhalt and Saxon-Thuringian horses. Females had the lowest average CPL-score and males the highest. ORs of males compared to females and geldings indicated a 5.08- and 1.92-fold higher risk. The average CPL-score was lowest in black horses and highest in bay horses. Bay horses exhibited ORs of 1.77 and 2.19 compared to chestnut and black horses, respectively. 

### 3.4. Testing Stud-Related Factors for CPL-Scores Using Generalized Linear Multivariable Models

We first tested each individual stud-related factor as an additional factor with model 1 (Appendix A) before starting a forward and backward selection of factors that were significant as individual factors. The final model 2 revealed 6/15 stud-related factors as significant in generalised linear models (Table 7). Outcomes from using a normal distribution did not differ from using a multinomial distribution. Interactions between sex or age within sex and any of the stud-related factors were not significant. Estimates of LS-means and ORs for stud-related variates are given in Appendix A. Horses kept on pastures or pastures and paddocks had lower risk to higher CPL-scores. Using rye straw combined with wheat straw as bedding type was deemed to increase the risk for higher CPL-scores. Shorter time intervals for cleaning out the stable decreased the risk for CPL-scores. Feeding hay with straw in winter lowered the risk for CPL-scores, whereas hay silage increased the risk. Feeding of concentrate in winter was positively related with higher CPL-scores. Type of concentrate was not significant and, therefore, all types of concentrates were regarded as one level. Hoof trimming intervals of about 16 weeks were associated with the lowest CPL-scores. Residual variances were found in models 1 and 2 for the CPL-score 0.8380 ± 0.05415 and 0.7405 ± 0.04899, resulting in a reduction of 11.6% (0.0975) in the residual variance in model 1 due to horse farm-related effects. Extending model 2 with the random effect for horse farm, the residual variance was reduced by a further 2.9% to a value of 0.7192. 

### 3.5. Testing Animal-Related Variables for CPL-Scores Using Generalized Linear Multivariable Models

The means and standard deviations of the animal-related variables are shown in Appendix A. Testing the animal-related variables as covariates revealed that cannon bone circumference was significant at the front left limb and right hind limb in generalised linear models (Appendix A). Height at withers reached the 0.05 significance level only in the cubic regression on CPL-scores but not in the linear and quadratic regressions. The cubic regression was not significant for the cannon bone circumference at the right hind limb and was therefore not included in the model. Estimates of the regression coefficients for the cannon bone circumference are given for all horses in which the cannon bone circumference was recorded and that were older than 1, 3, 6 or 9 years (Table 8). Increases in CPL-score per 1 cm of cannon bone circumference at the front and hind limb was 2.12-fold (CI: 1.84–2.44) and 2.04-fold (CI: 1.77–2.32), respectively. The corresponding Ors for males, geldings, and females at the front limb were 2.35 (CI: 1.88–2.94), 2.54 (CI: 1.73–3.25), and 2.04 (CI: 1.76–2.37), respectively, and at the hind limb 1.79 (CI: 1.40–2.28), 2.36 (CI: 1.68–3.31), and 2.08 (CI: 1.77–2.46), respectively. However, these differences between sexes were not significant. 

## 4. Discussion

In this study, we analysed the prevalence and progression of CPL with age and evaluated animal- and horse farm-related factors influencing the prevalence and severity of CPL-lesions in Rhenish German draught horses. We employed multivariable generalised linear models to identify risk factors for CPL-scores. Due to the large sample size and large number of horse studs, factors under analysis were not confounded. Participation in this study was voluntary, and breeders were required to agree for taking part in this study. Thus, our study was not completely randomized. However, confidence intervals are reasonably small for the prevalence of CPL, and therefore support the robustness of our results.

The classification of CPL took into account the various progressing forms and severity of CPL [1,2,3,12,13,25]. All recordings were done on horse farms in order to sample all horses per stud and in particular the older horses that were no longer used for breeding. For this reason, we did not visit horse registration or horse breeding events or horse contests for recording, as was the case in a previous Belgian study [2,23]. The proportion of horses older than 3 years was lower in the previous Belgian study, 52.6% [23], than in the present study, 63.5%. Mares at stable visits were significantly older than mares on horse contests [2]. In addition, horses sampled on horse contests had significantly lower CPL prevalence (54% vs. 66%) and lower CPL-scores in mares (2.49 vs. 5.34) than horses examined at stable visits [2]. 

Comparison of the prevalence of CPL between different studies is difficult due to study design, age distribution of horses sampled, and type of sampling. The age when horses are sampled has a major effect on prevalence, as CPL progresses significantly with age [1,2,10,12,23,25]. Often, it is not possible to sample most horses for a certain breed because recording is not mandatory for a breeding program and breeders can decide whether to participate in a study. In the Belgian study [23], 56% in the full data set with 762 horses and 88% in the dataset with 401 horses older than 3 years were affected by CPL. The mean age in the full data set of the Belgian study was 3.87 ± 3.02 years [2]. In the previous German study [1,10,25], horses older than 2.5 years were recorded, and the prevalence estimated in a model correcting for systematic effects such as age and sex ranged widely from 24% to 96%. Estimates were between 75 and 96% for breeds with a genealogical relationship with Belgian draught horses [29], and lowest for Black Forest at 24%. Overall, the prevalence of CPL in Belgian draught horses [2,23] appear to be similar to those in the Rhenish-German draught horses in the present study and slightly lower than in the previous German study [1,25]. The horses had a higher mean age (8.50 ± 4.38 and 7.77 ± 6.37 years) in both German studies, the previous [1,25] and the present study, which may contribute to the similar prevalence compared with the Belgian study [2,23]. 

All studies agree that males or stallions are more frequently affected by CPL [1,2,3,4,5,6,7,8,9,10,11,12,13,14,15,16,17,18,19,20,21,22,23,24,25]. The higher susceptibility of males developing more severe CPL-lesions is believed to be a result of the stabling and management of breeding stallions and the breeding aim of large-framed stallions with strong cannon bones in previous times [10,30]. 

An increase in CPL prevalence with age was found in Belgian [2,23] and German draught horse breeds [1,10,25]. In the present study, similar to the Belgian study [2,23], we confirmed a significant interaction of sex and age, whereas in the previous studies [1,10,25], an interaction of sex and age was not evident and only a linear increase with age was found. In addition, the increase in CPL prevalence was significantly steeper in Rhenish German, Mecklenburg and Saxon-Thuringian draught horses than in Black Forest and South German [1]. Disease progression is significantly faster in males, which is in agreement with the Belgian study [2,23], but not with previous German studies [1,10,12,25]. As the clinical signs of CPL progress in older horses, lesions become more painful and interfere with movement, the intervals between recurrent infections become shorter, and the horses lose appetite and their condition deteriorates, which may result in the need to euthanize the horses. Therefore, severely CPL-affected horses may not reach the same age as only mildly affected horses, so the proportion of severely CPL-affected horses may decrease at older ages. An explanation for these differences in disease progression between males and females is thought to be differences in the type of housing and less access to pastures [25,26,27] as well as more intensive selective breeding in stallions [7,25,30].

Differences between breeding associations (regions) of the Rhenish German draught horse were found in the previous [1,25] and the present German study. A possible reason for these differences could be genetic differences between these subpopulations [29]. There seems to be some exchange of breeding horses between these regions, but far from a complete mixing of breeding populations. Even when the model used accounted for interactions between age and sex and coat colour and was extended to include horse farm-related effects, the significant differences between breeding associations remained. Differences in age structure among these breeding associations were not important because we did not detect interactions between age and breeding associations or between breeding association, sex, and age. We also examined the different types of use and work assignments of the horses, but could not demonstrate a significant contribution to higher CPL-scores. Draught horses in eastern Germany were more frequently used in agriculture and dairy production in the past, which might have led to a stricter selection against CPL.

The effects of coat colour do not seem to be consistent among the different previous studies [2,23,25] and may be partly associated with sires or horse families. However, the association of increased CPL prevalence with chestnut and bay Rhenish German horses in Saxony-Anhalt was also stated in an early German study [31]. A direct effect of horse colour variants on the occurrence of CPL has not yet been considered. Significant quantitative trait loci for CPL were not located on the horse chromosomes where coat colour genes (*ASIP* on ECA 22, *MC1R* on ECA 3, *SLC45A2* on ECA 21 and *KIT* on ECA 3) have been mapped [28]. Even larger samples seem to be necessary to reach a clearer conclusion.

In agreement with previous studies, we confirmed the negative effects of some feeding rations [1,12,25,26,27]. Feeding of concentrates and hay silage are factors promoting CPL [1,25]. Access to pasture and improvement in stable hygiene help to reduce the prevalence and progression of CPL [1,12,25]. Optimization of housing conditions, feeding rations and management may have positive effects on preventing disease occurrence and progression, but these environmental effects could not cancel out the effects of the breeding area, sex, and age by sex interactions. Thus, these horse farm-related effects explained only a part of the total variation. Exercise is believed to enhance lymphatic flow and therefore supports the transportation of lymph fluids [19,32]. Horses used for riding, carriage, and work in agriculture were less prone to severe CPL-lesions when the type of work application was added as a single factor to model 1. Therefore, it is important to improve opportunities for exercise in draught horses. Even though this effect was not significant in the final model 2, and only a trend in this direction was observed.

We studied animal variables in order to prove whether body traits might show correlated changes with CPL and thus be influenced by the impaired lymphatic flow [11,14,15,16]. The significant positive correlation between CPL-scores and cannon bone circumference is consistent with previous studies [7,12,30]. The significant correlations with cannon bone circumference at the front and hind limb were still present even if we removed age and sex effects for cannon bone circumference in the analysis. This result means that larger cannon bone circumferences are correlated with CPL-scores, independent of the age and sex of the horses. This might indicate a genetic disposition to more severe CPL-lesions in horses with larger cannon bone circumference. Height at withers was not significant in our study, therefore we cannot confirm that large-framed horses are significantly more likely affected by CPL-lesions [26,30]. Similarly, hoof measures and hoof horn hardness do not seem to be influenced by CPL and its underlying stagnating lymphatic flow.

## 5. Conclusions

The results of this study demonstrate an overall high prevalence of CPL in Rhenish German draught horse, but also significant differences in CPL prevalence and CPL-scores among the different breeding associations. This may indicate genetic differences due to genealogies in the West and East German breeding lines. Because it takes a long time for lesions to become clearly visible to the horse owner, careful screening at an early age is very important to start medical interventions as early as possible to reduce the severity of lesions and to prolong the life expectancy of CPL-affected horses. Reducing horse farm-related risk factors may have significant effects on the occurrence and progression of CPL. Further studies are warranted to investigate genetic disposition and to develop procedures to estimate breeding values in the Rhenish German population. 

## Figures and Tables

**Figure 1 animals-13-00999-f001:**
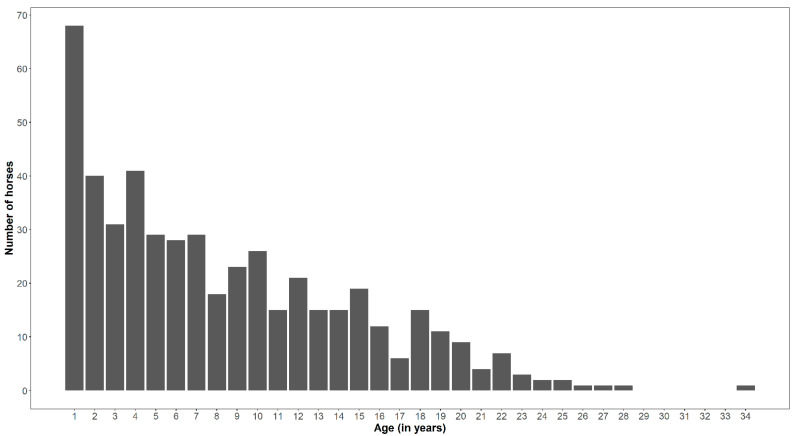
Distribution of the 493 sampled Rhenish German Draught horses by age in years including foals below one year old (category = 1) and up to a 33 years old horse (category = 34).

**Table 1 animals-13-00999-t001:** Number of sampled Rhenish German draught horses by region, horse breeding association and sex (n = 493).

Region (Breeding Association)	Studs	Horses	Proportion by Sex (Male/Gelding/Female)		
Westphalia (Westphalian)	75	268	0.21	0.15	0.64
Thuringia (Saxon-Thuringian)	3	106	0.17	0.03	0.80
Brandenburg (Brandenburg-Anhalt)	2	67	0.40	0.08	0.52
Rhineland (Rhenish)	14	43	0.12	0.19	0.70
Lower Saxony (Lower Saxon)	2	9	0.67	0.00	0.33
Total	96	493	0.23	0.12	0.66

**Table 2 animals-13-00999-t002:** Percentage and 95% confidence intervals (CI) of horses by scores of chronic progressive lymphedema (CPL) and age groups in 493 Rhenish German draught horses.

Age Group (Years)	No of Horses	CPL-Score						Overall CPL- Score (Mean ± SD)
		0	1	2	3	4	5	
<1	68	40.0	0.0	0.0	0.0	0.0	0.0	0.0
1–3	71	27.0	29.0	12.9	2.5	0.0	0.0	0.648 ± 0.927
3–6	98	20.6	19.4	34.7	18.3	0.0	0.0	1.337 ± 1.130
6–9	70	5.9	22.6	20.8	17.5	16.9	0.0	2.114 ± 1.325
9–12	62	2.4	19.4	12.9	20.8	18.5	33.3	2.516 ± 1.127
12–18	82	3.5	6.5	13.9	23.3	44.6	50.0	2.805 ± 1.222
>17	42	0.6	3.2	5.0	17.5	20.0	16.7	2.976 ± 0.999
Total (n)	493	170	31	101	120	65	6	1.696 ± 1.464
Percentage	100	34.5	6.3	20.5	24.3	13.2	1.2	
CI		30.3–38.7	4.2–8.8	16.9–24.1	20.1–28.1	10.2–16.2	0.1–2.2	

**Table 3 animals-13-00999-t003:** Numbers and percentages with 95% confidence intervals (CI) of males, females and geldings by chronic progressive lymphedema scores in 493 Rhenish German draught horses.

CPL-Score	No of Males	Percentage (CI)	No of Females	Percentage (CI)	No of Geldings	Percentage (CI)
0	48	43.2 (34.1–52.5)	117	36.0 (30.7–41.2)	5	8.8 (3.0–19.6)
1	5	4.5 (1.5–10.2)	23	7.1 (4.5–10.4)	3	5.3 (1.1–14.6)
2	19	17.1 (10.6–25.4)	69	21.2 (16.9–26.1)	13	21.8 (12.7–35.8)
3	19	17.1 (10.6–25.4)	76	23.4 (18.9–28.4)	25	43.9 (30.7–57.6)
4	16	14.4 (8.5–22.4)	39	12.0 (8.7–16.0)	10	17.5 (8.8–29.9)
5	4	3.6 (1.0–9.0)	1	0.3 (0–1.7)	1	1.8 (0–9.4)
Total	111	100	325	100	57	100
CPL-score (average ± SD)	1.577 ± 1.593		1.603 ± 1.429		2.456 ± 1.166	

**Table 4 animals-13-00999-t004:** Results of the generalized linear mixed model 1 with degrees of freedom (DF), F-values and *p*-values for CPL-scores in Rhenish German draught horses.

Source of Variation	DF	Normal Distribution		Multinomial Distribution	
		F-Value	*p*-Value	F-Value	*p*-Value
Breeding association	3	21.31	<0.0001	21.31	<0.0001
Sex	2	5.84	0.0031	6.48	0.0017
Coat colour	2	3.06	0.0479	4.89	0.0079
Age (linear) by sex	3	92.87	<0.0001	56.99	<0.0001
Age (quadratic) by sex	3	33.83	<0.0001	29.79	<0.0001

**Table 5 animals-13-00999-t005:** Estimates of linear and quadratic regression coefficients for age by sex with their standard errors (SE) and *p*-values for CPL-scores in all Rhenish German draught horses and with an age greater equal than 1, 2, 3, 4, 5, 6, 7 and 8 years, using model 1 with a normal distribution function.

Age	Regression Coefficients with Their Standard Errors			*p*-Value
	Male	Gelding	Female	
All horses (n = 493)				
- linear	0.5795 ± 0.0523	0.1416 ± 0.0785	0.3146 ± 0.0254	<0.0001
- quadratic	−0.0187 ± 0.0028	−0.0023 ± 0.0030	−0.0083 ± 0.0011	<0.0001
Horses ≥ 1 year (n = 425)				
- linear	0.5418 ± 0.0705	0.1360 ± 0.0819	0.3242 ± 0.0306	<0.0001
- quadratic	−0.0171 ± 0.0034	−0.0020 ± 0.0031	−0.0086 ± 0.0013	<0.0001
Horses ≥ 2 years (n = 385)				
- linear	0.5255 ± 0.0917	0.1566 ± 0.0879	0.3029 ± 0.0364	<0.0001
- quadratic	−0.0164 ± 0.0042	−0.0027 ± 0.0033	−0.0079 ± 0.0014	<0.0001
Horses ≥ 3 years (n = 354)				
- linear	0.4627 ± 0.1122	0.1633 ± 0.0882	0.3229 ± 0.0419	<0.0001
- quadratic	−0.0140 ± 0.0048	−0.0030 ± 0.0033	−0.0086 ± 0.0016	<0.0001
Horses ≥ 4 years (n = 313)				
- linear	0.4539 ± 0.1417	0.1126 ± 0.1069	0.2795 ± 0.0502	<0.0001
- quadratic	−0.0136 ± 0.0057	−0.0014 ± 0.0038	−0.0073 ± 0.0018	<0.0001
Horses ≥ 5 years (n = 284)				
- linear	0.4600 ± 0.1998	0.1179 ± 0.1265	0.2916 ± 0.0577	<0.0001
- quadratic	−0.0137 ± 0.0078	−0.0016 ± 0.0043	−0.0076 ± 0.0020	0.0006
Horses ≥ 6 years (n = 256)				
- linear	0.3829 ± 0.2454	0.2088 ± 0.1592	0.2793 ± 0.0664	0.0001
- quadratic	−0.01134 ± 0.0089	−0.0041 ± 0.0051	−0.0073 ± 0.0022	0.0054
Horses ≥ 7 years (n = 227)				
- linear	0.4051 ± 0.2946	0.2061 ± 0.1707	0.2540 ± 0.0829	0.0063
- quadratic	−0.0122 ± 0.0104	−0.0042 ± 0.0053	−0.0066 ± 0.0026	0.0394
Horses ≥ 8 years (n = 209)				
- linear	0.5193 ± 0.4228	0.2649 ± 0.1878	0.2189 ± 0.0914	0.0295
- quadratic	−0.0160 ± 0.0141	−0.0056 ± 0.0057	−0.0056 ± 0.0028	0.1040

**Table 6 animals-13-00999-t006:** Least square mean estimates (LSM) and odds ratios (OR) for the fixed effects of horse breeding association (HBA), sex and coat colour with their standard errors (SE) and 95% confidence intervals (95%-CI) and *p*-values for CPL-scores (cumulative logit estimates = C-Log or LSM differences) in all Rhenish German draught horses using model 1 with a normal and multinomial distribution function.

Effect/ Level	Level	LSM ± SE	*p*-Values	OR (95%-CI)	*p*-Values
			LSM		C-Log
HBA					
	Brandenburg-Anhalt	1.51 ± 0.12	0.01/0.61/0.001	8.16 (4.25–15.67)	0.001/0.43/0.001
	Rhenish	2.06 ± 0.16	-/0.01/0.06	1.87 (0.97–3.61)	-/0.003/0.06
	Saxon-Thuringian	1.59 ± 0.13	-/-/0.001	6.05 (3.47–10.57)	-/-/0.001
	Westphalian	2.36 ± 0.08	-	-	-
Sex					
	Female	1.45 ± 0.07	0.001/0.001	5.08 (2.99–8.63)	0.006/0.001
	Gelding	1.97 ± 0.16	-/0.188	1.92 (0.94–3.95)	-/0.07
	Male	1.99 ± 0.11	-	-	-
Coat colour					
	Chestnut	1.84 ± 0.10	0.6303/0.0376	1.77 (1.16–2.71)	0.57/0.008
	Black	1.76 ± 0.16	-/0.0828	2.19 (1.08–4.45)	-/0.03
	Bay	2.04 ± 0.08	-	-	-

*p*-Values: differences between C-Log and LSM to other levels are given in each line, sorted by the first to the last level per effect.

**Table 7 animals-13-00999-t007:** Results of the final generalized linear mixed model 2 including horse farm-related factors in addition to model 1 with degrees of freedom (DF), F-values and *p*-values for CPL-scores in Rhenish German draught horses.

Source of Variation with Horse Farm-Related Factors	DF	Normal Distribution		Multinomial Distribution	
		F-Value	*p*-Value	F-Value	*p*-Value
Breeding association	3	12.44	<0.0001	10.16	<0.0001
Sex	2	7.90	0.0004	7.28	0.0008
Coat colour	2	4.81	0.0090	2.96	0.0530
Age (linear) by sex	3	49.49	<0.0001	74.11	<0.0001
Age (quadratic) by sex	3	24.54	<0.0001	26.82	<0.0001
Outdoor facilities for horses in summer (OUTS)	2	4.06	0.0073	4.22	0.0059
Bedding type (BED)	7	3.37	0.0016	2.37	0.0221
Time interval for cleaning out the stable (CLEAN)	3	8.22	<0.0001	6.60	0.0002
Type of roughage fed in winter months (ROUW)	4	2.54	0.0394	3.28	0.0115
Type of concentrate fed in winter months (CONW)	1	9.65	0.0020	7.53	0.0063
Length of hoof trimming intervals (HOFT)	4	4.16	0.0026	5.33	0.0003

**Table 8 animals-13-00999-t008:** Estimates of linear, quadratic and cubic regression coefficients with their standard errors (SE) and *p*-values of the cannon bone circumference at the front left limb and right hind limb on CPL-scores for Rhenish German draught horses and with ages greater than 1, 3, 6, and 9 years using a normal distribution function.

Age	Regression Coefficients with Their Standard Errors and *p*-Values			
	Front Limb	*p*-Value	Hind Limb	*p*-Value
Horses ≥ 1 year (n = 396)				
- linear	2.8298 ± 0.8884	0.0016	0.9506 ± 0.2368	<0.0001
- quadratic	−0.07092 ± 0.02838	0.0129	−0.01157 ± 0.003839	0.0028
- cubic	0.000574 ± 0.000293	0.0506		
Horses ≥ 3 years (n = 336)				
- linear	6.8489 ± 1.1279	<0.0001	1.6400 ± 0.2869	<0.0001
- quadratic	−0.1946 ± 0.03596	<0.0001	−0.02225 ± 0.004575	<0.0001
- cubic	0.001807 ± 0.000372	<0.0001		
Horses ≥ 6 years (n = 243)				
- linear	8.0452 ± 1.2289	<0.0001	1.9710 ± 0.3001	<0.0001
- quadratic	−0.2304 ± 0.03888	<0.0001	−0.02706 ± 0.004732	<0.0001
- cubic	0.002155 ± 0.000400	<0.0001		
Horses ≥ 9 years (n = 176)				
- linear	7.8459 ± 1.3991	<0.0001	12.4234 ± 3.9350	0.0019
- quadratic	−0.2248 ± 0.04413	<0.0001	−0.3488 ± 0.1220	0.0048
- cubic	0.002109 ± 0.000452	<0.0001	0.003279 ± 0.001251	0.0096

## Data Availability

Restrictions apply to the availability of these data. Data were obtained from German horse owners, breeders, and the Rhenish and Westphalian breeder associations and are available from the authors at a reasonable request and with the permission of the horse owners.

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
