# Peer review of "Prevalence of Chronic Progressive Lymphedema in the Rhenish German Draught Horse"

_animals, 2023, doi:10.3390/ani13060999_

Round 1

Reviewer 1 Report

General: This is a very extensive and thorough study. However, many of the results are not new information. I think the manuscript needs some adjustments and shortening before being accepted for publication. There are some terms that are not clear (please also see below). Please have it evaluated for grammar and vocabulary before resubmitting.

A few issues throughout the manuscript:

-       What do you mean by “limb position” (example line 27)

-       The frequency of coat color of affected horses needs to be correlated with frequency of that coat color within the breed to make a point that certain coat colors may be associated with higher incidence of CPL

-       The category ”sexes by age” is never really clearly defined. You independently list the progression with age as well as higher incidence in stallions.

-       Use the term “stallion” instead of males (as you also use geldings and mares and not females and castrated males)

-       Geldings are the least affected. This may not be surprising, as stud firms usually do not keep geldings? Can you specify the type of farms you were investigating.

A few specific things noted:

51-52: the breeds need to be listed in plural

71-72: fibrosis does not affect the actual vascular walls

79: defects of elastin do NOT interfere but are associated with

112: use management instead of “horse farm factors”; why not using “farm management”

202: Litter not usually used for manure and bedding

218: right hind

Table 8: What do you mean by “dunging interval”

Many areas of the manuscript should be shortened:

99-101: could be shortened, “As a progressive behavior of the disease, lesions worsen with age!

Discussion repeats a lot of results – this is cumbersome! Try to be more descriptive and summarizing results.

Author Response

Reviewer 1:

We thank the reviewer and editor for his time and labor to improve our manuscript as well as for the valuable comments and recommendations.

We addressed all comments of the reviewer.

General: This is a very extensive and thorough study. However, many of the results are not new information. I think the manuscript needs some adjustments and shortening before being accepted for publication. There are some terms that are not clear (please also see below). Please have it evaluated for grammar and vocabulary before resubmitting.

A few issues throughout the manuscript:

-       What do you mean by “limb position” (example line 27)

Amended

Line 28: limb on which CPL was recorded,

Line 278-280: CPL-scores and prevalence were not significant between limbs, and thus, CPL-scores for the single limbs were not used in all following analyses.

-       The frequency of coat color of affected horses needs to be correlated with frequency of that coat color within the breed to make a point that certain coat colors may be associated with higher incidence of CPL

Amended: We added a Supplementary Table S3 to relate these frequencies. In addition, we analysed coat color as a fixed effect in multivariable models and also tested all two-way-interactions between coat color and breeding association, sex and age at examination.

Lines

-       The category ”sexes by age” is never really clearly defined. You independently list the progression with age as well as higher incidence in stallions.

Amended:

We employed linear and quadratic regressions on age within the three different sexes. Thus we obtained each 3 linear and 3 quadratic regression coefficients, each one per sex.

Lines 275-277: b1(age*sex)l  = linear regression on age in years within the three different sexes; b2(age2 *sex)m = quadratic regression on age in years within the three different sexes;

In Table 4: Age (linear) by sex, Age (quadratic) by sex

-       Use the term “stallion” instead of males (as you also use geldings and mares and not females and castrated males)

Comment: stallions deems not correct because we also have male horses aged 1 or 2 years. We use only the term mare when citing literature and authors are referring to mares. The occasions where we used the term stallion and mare have been corrected.

Castrated males are males that are gelded. Seems no difference. Castrated is more common in veterinary medicine.

-       Geldings are the least affected. This may not be surprising, as stud firms usually do not keep geldings? Can you specify the type of farms you were investigating.

Amended:

Type of farm:

All owners of the studs visited are horse breeders. Breeders of draught horses keep also geldings for working purposes because they use horses for wagon rides, traditional costume parades (e.g., the very known Munich Octoberfest) and for sale. Type of use and work application of the horses should characterize the type of farm. This variable was investigated but did not have an effect on CPL-score. We did not visit farms where horse are kept as a hobby or for pleasure. Geldings are mainly used for wagon rides and parades.

Amended:

Lines 172-179:

We invited all Rhenish German Draught horse breeders in North Rhine-Westphalia to participate in the study. In addition, we offered breeders from other German states to be involved in this project. All breeders that agreed to participate in this project got an information letter and were contacted to arrange a visiting appointment for examination of the horses and recording management, housing and feeding data of the horses. All breeders recruited for this study participated voluntarily and gave written informed consent for participating in this research on CPL and possible risk factors.

Line 207-213:

During each horse farm visit, an interview with the horse owner was conducted. The stud was inspected to record data on the type of stable, the type of use or work applications for the horses, the type of bedding, manure management, type of outdoor access and average hours of outdoor exercise per day in autumn and winter or in spring and summer, average hours of pasture grazing, frequency of shoeing, type of roughage and concentrates fed in summer and winter months. The answers were documented on a written questionnaire (Supplementary Table S1).

A few specific things noted:

51-52: the breeds need to be listed in plural

Amended:

Lines 53-54: Shires (n=28), Saxon-Thuringians (n=20), Belgian Draught horses (n=10), Clydesdales (n=3), Rhenish German horses (n=2), and Percherons (n=1).

71-72: fibrosis does not affect the actual vascular walls

Amended:

Line 44: fibrosis of dermis and subcutaneous tissues, and fragmentation and disorganization of elastic fibers

79: defects of elastin do NOT interfere but are associated with

Amended:

Lines 80-81: Defects of the network of elastic fibers surrounding dermal lymphatics are assumed to be associated with chronic lymphedema

112: use management instead of “horse farm factors”; why not using “farm management”

Comment: we did not change into „farm management“ because we were looking not just on management but also on type of stable, feeding rations, working with horses.

202: Litter not usually used for manure and bedding

Amended:

Line 209, 293, 414-415

Bedding type

Manure management

218: right hind

Amended:

Line 225: from the left front and right hind limb.

Table 8: What do you mean by “dunging interval”

Amended:

Line 293, 415-416: time interval for cleaning out the stable

Many areas of the manuscript should be shortened:

99-101: could be shortened, “As a progressive behavior of the disease, lesions worsen with age!

Amended:

Line 101-102: Occurrence and severity of CPL-lesions showed interactions between sex and age in previous studies

Discussion repeats a lot of results – this is cumbersome! Try to be more descriptive and summarizing results.

Amended:

Lines 482-487: removed.

Lines 504-508: removed.

Lines 470-482: in this part, there is a number of results from previous studies and the present study. We refer to these results to underline the hypotheses and issues raised. Otherwise, this part would be very general and it would be hard to workout the differences among studies and breeds. 

Reviewer 2 Report

The paper entitled "Prevalence of chronic progressive lymphedema in the Rhenish German Draught Horse" it is relevant and provides knowledge on the prevalence of CPL in this breed and the factors influencing it. However, I do have some suggestions that might improve it.

The introduction gives an extensive overview of all literature available on CPL in draught horses, not only in German horses, including the breed under study, but also in other breeds.  The introduction concludes with "Therefore, the objective of this study was to examine the current prevalence .... in Rhenish German Draught horses in Germany", but it could have been explained better why this breed specifically was chosen and not the other German ones mentioned in the introduction. 

References are not always correct, e.g. line 136 should be reference 4, and line 106 reference 11, according to me. 

The study is based on farm visits including examination of the horses and recording of management, housing and feeding data of the horses based on a questionnaire. It is not clear if all examinations and questionnaires have been done by the same person, and if not, if there was an effect of the examinator (line 175).

In the results section the references to the tables do not match with the table numbers (eg;. line 300 should be Table 2; line 317 should be Table 3) Table 5 is missing.

According to me Table 3 is less relevant because CPL score largely depends on the age and that is not the same for the 3 sexes. Figures S2 till S4 are more informative. Legends of figures are very small and it would be good to enlarge them. 

Figures S5-S7 are difficult to interprete, what exactly is this proportion of CPL? It is not clear from the text, nor from the figure. line 331 states that in males of 6-9 years (age group 4 in S5 it is 6-8 years?) 75% of the males have CPL scores >2. But what is then the blue line on 100% on age 4? Please explain this better.

A lot of results are given and it is not always clear which conclusions can be drawn from which tables. In the discussion the effect of the different factors becomes more clear. However, if there is a significant effect of a certain variable, it would be good to specify the magnitude of the effect, where relevant.

Line 518, the effect of excercise as found in literature is discussed, but what was the effect in this study?

In the conclusion it is stated that screening at an early age is very important to start medical interventions as early as possible. How do you deduct that from this study?

Author Response

Reviewer 2:

We thank the reviewer and editor for his time and labor to improve our manuscript as well as for the valuable comments and recommendations.

We addressed all comments of the reviewer.

The paper entitled "Prevalence of chronic progressive lymphedema in the Rhenish German Draught Horse" it is relevant and provides knowledge on the prevalence of CPL in this breed and the factors influencing it. However, I do have some suggestions that might improve it.

The introduction gives an extensive overview of all literature available on CPL in draught horses, not only in German horses, including the breed under study, but also in other breeds.  The introduction concludes with "Therefore, the objective of this study was to examine the current prevalence .... in Rhenish German Draught horses in Germany", but it could have been explained better why this breed specifically was chosen and not the other German ones mentioned in the introduction. 

Amended:

Lines 141-145

Previous reports indicated that CPL may have a significant impact on the health of Rhenish German draught horses [7,25] and therefore breeders may be discouraged from keeping this breed. In this way, CPL could be a reason why the population size of the Rhenish German draught horse population continues to decrease below 1000 breeding animals.

References are not always correct, e.g. line 136 should be reference 4, and line 106 reference 11, according to me. 

Amended:

Line 105-106:

Both studies with 431 and 980 records of Belgian draught horses confirmed significant age and sex by age effects on CPL severity [2,24].

Line 135-136:

horses with 26 cases (CPL-affected) and 19 controls did not identify significantly associated loci for CPL [4].

The study is based on farm visits including examination of the horses and recording of management, housing and feeding data of the horses based on a questionnaire. It is not clear if all examinations and questionnaires have been done by the same person, and if not, if there was an effect of the examinator (line 175).

Amended:

All examinations have been done by one person. There is no effect of the examiner.

Line 240-241: There was no influence of the investigator to be considered, as all examinations were performed by an experienced veterinarian.

In the results section the references to the tables do not match with the table numbers (eg;. line 300 should be Table 2; line 317 should be Table 3) Table 5 is missing.

Amended: We checked all numberings of the Tables and amended the errors in the table numbers.

According to me Table 3 is less relevant because CPL score largely depends on the age and that is not the same for the 3 sexes. Figures S2 till S4 are more informative. Legends of figures are very small and it would be good to enlarge them. 

Comments: We kept Table 3 because sex is significant as a sole factor. We kept Figures S2-S4 in the supplementary section because the figures are very large.

Amended:

Legends were enlarged.

Figures S5-S7 are difficult to interprete, what exactly is this proportion of CPL? It is not clear from the text, nor from the figure. line 331 states that in males of 6-9 years (age group 4 in S5 it is 6-8 years?) 75% of the males have CPL scores >2. But what is then the blue line on 100% on age 4? Please explain this better.

Amended:

Lines 342-347:

The proportions of horses with CPL-scores >0, >1, >2, and >3 within age classes are given for males, geldings and females in Supplementary Figures S5-S7. In males, the steepest increase in the proportion of horses with CPL-scores >2 and >3 is observed between ages 3-5 (age group 3) and 6-8 (age group 4) years. In female horses, the increase in the proportion of horses with CPL-scores >2 and >3 is much flatter and extends over a longer age.

Figure S5:

The blue line represents the proportion of male horses in the respective age group with CPL-scores 1-5, the red line the proportion with CPL-scores 2-5, the dark green line the proportion with CPL-scores 3-5, and the brown line the proportion with CPL-scores 4-5.

Correspondingly for Figure S6 and S7.

A lot of results are given and it is not always clear which conclusions can be drawn from which tables. In the discussion the effect of the different factors becomes more clear. However, if there is a significant effect of a certain variable, it would be good to specify the magnitude of the effect, where relevant.

Amended:

We specified significant effects and pointed out the size of the effects.

Line 390-397:

Horses of the horse breeding association from Brandenburg-Anhalt showed the lowest average CPL-score, whereas Westphalian horses reached the highest average CPL-score (Table 6). The ORs of Westphalian horses for CPL-scores was 8.16 and 6.05 times higher compared to Brandenburg-Anhalt and Saxon-Thuringian horses. Females had the lowest average CPL-score and males the highest. ORs of males compared to females and geldings indicated a 5.08- and 1.92-fold higher risk. The average CPL-score was lowest in black horses and highest in bay horses. Bay horses exhibited ORs of 1.77 and 2.19 compared to chestnut and black horses, respectively.

Line 413-420:

Horses kept on pastures or pastures and paddocks had lower risk to higher CPL-scores. Using rye straw combined with wheat straw as bedding type deemed to increase risk for higher CPL-scores. Shorter time intervals for cleaning out the stable decreased risk for CPL-scores. Feeding hay with straw in winter lowered risk for CPL-scores, whereas hay silage increased risk. Feeding of concentrate in winter was positively related with higher CPL-scores. Type of concentrate was not significant and therefore, all types of concentrates were regarded as one level. Hoof trimming intervals of about 16 weeks were associated with the lowest CPL-scores.

Line 518, the effect of excercise as found in literature is discussed, but what was the effect in this study?

Amended:

Lines 546-550:

Horses used for riding, carriage and work in agriculture were less prone to severe CPL-lesions when the type of work application was added as a single factor to model 1. Therefore, it is important to improve opportunities for exercise in draught horses. Even though this effect was not significant in the final model 2, and only a trend in this direction was observed.

In the conclusion it is stated that screening at an early age is very important to start medical interventions as early as possible. How do you deduct that from this study?

Amended:

Lines 567-568

Because it takes a long time for lesions to become clearly visible to the horse owner, careful screening at an early age is very important to start medical interventions as early as possible to reduce the severity of lesions and to prolong life expectancy of CPL-affected horses.

Reviewer 3 Report

REVIEW

“Prevalence of Chronic Progressive Lymphedema in the Rhenish German Draught Horse”

BRIEF SUMMARY AND GENERAL COMMENTS:

The present manuscript aims to describe the prevalence of Chronic Progressive Lymphemeda (CPL) in Rhenish German Horses and to identify risk factors for presence and severity of lesions. a Good bibliographical review has been performed, and the introduction and discussion sections are clear and exhaustive. The statistical approach used in the present study is very complex, which can make the paper not so reader friendly. However, I think it is appropriate for a large epidemiological study which aims to provide accurate results. Overall, the quality of the manuscript is high, and English is good. Please find some minor comments below.

SPECIFIC COMMENTS:

Simple summary

Line 10: please remove “with age”, as it is a repetition.

Introduction

Lines 49-51: I suggest rephrasing this sentence to make it more readable, i.e. “In a review on the causes of premature leave of 979 coldblooded stallion from the stud Kreuz, Germany, the CPL was the primary…”

Lines 51-52: I suggest rephrasing this sentence as follows: “Stallions belonged to different breeds, including Shire…”

Lines 69-78 and 80-91: The pathological findings should be reported in the present tense, similarly to the rest of the introduction.

Line 147: “as well as”

Materials and Methods

Line 165: “...presented the project TO the breeders…”

Lines 176-199: These data should be moved in the Results section. In particular, I suggest adding a subsection including the signalment data of enrolled horses.

Line 187: A “%” should be added after 15.8.

Line 216: The correct verb is “were” as there are two subjects.

Lines 219-220: Why did you choose the left front and right hind hooves?

Author Response

Reviewer 3:

We thank the reviewer and editor for his time and labor to improve our manuscript as well as for the valuable comments and recommendations.

We addressed all comments of the reviewer.

BRIEF SUMMARY AND GENERAL COMMENTS:

The present manuscript aims to describe the prevalence of Chronic Progressive Lymphemeda (CPL) in Rhenish German Horses and to identify risk factors for presence and severity of lesions. a Good bibliographical review has been performed, and the introduction and discussion sections are clear and exhaustive. The statistical approach used in the present study is very complex, which can make the paper not so reader friendly. However, I think it is appropriate for a large epidemiological study which aims to provide accurate results. Overall, the quality of the manuscript is high, and English is good. Please find some minor comments below.

SPECIFIC COMMENTS:

Simple summary

Line 10: please remove “with age”, as it is a repetition.

Line 10:

females in Rhenish German draught horses.

Introduction

Lines 49-51: I suggest rephrasing this sentence to make it more readable, i.e. “In a review on the causes of premature leave of 979 coldblooded stallion from the stud Kreuz, Germany, the CPL was the primary…”

Amended:

Lines 50-52:

In a review on the causes of premature leave of 979 coldblooded stallions from the stud Kreuz, Germany, the CPL was the primary reason

Lines 53-54: I suggest rephrasing this sentence as follows: “Stallions belonged to different breeds, including Shire…”

Amended:

Lines 53-54:

Stallions belonged to different breeds, including Shire (n=28), Saxon-Thuringian (n=20), Belgian Draught (n=10), Clydesdale

Lines 69-78 and 80-91: The pathological findings should be reported in the present tense, similarly to the rest of the introduction.

Amended:

Lines 71-80: Pathological examinations of the distal limbs reveal epidermal hyperplasia,

Lines 82-93:

CPL-affected Clydesdale and Shire horses show higher cutaneous desmosine levels than clinically normal horses of the same breeds, but controls of the breed Percheron have significantly higher cutaneous desmosine levels than CPL-unaffected Clydesdale and Shire horses [15]. Mildly CPL-affected horses have the highest cutaneous desmosine levels, but levels decrease in more severely CPL-affected horses. Accumulation of elastin and amounts of desmosine are highest in superficial dermis in the distal limb and neck [16,17]. However, a study in Belgian draught horse stallions could not confirm the use of cutaneous desmosine levels as a diagnostic aid for CPL [18]. In summary, progressive lymphedema and tissue fibrosis along with disorganized elastic fibers supporting lymphatic vessels impair lymphatic clearance and correlate with disease progression, but the primary causative factors for the development of CPL remain still unknown [13-15]. 

Line 147: “as well as”

Amended:

Line 153: between CPL and animal as well as horse farm-related variables.

Materials and Methods

Line 165: “...presented the project TO the breeders…”

Amended:

Line 171: We presented the project to the breeders in several meetings

Lines 176-199: These data should be moved in the Results section. In particular, I suggest adding a subsection including the signalment data of enrolled horses.

Amended:

We moved these data on coat color to Results and added the subsection “3.2 Signalment data of enrolled horses”. We also an added Supplementary Table S3 on coat color distribution and CPL scores.

Line 187: A “%” should be added after 15.8.

Amended:

Line 193: only a few geldings (15.8%)

Line 216: The correct verb is “were” as there are two subjects.

Amended:

Line 223: were

Lines 219-220: Why did you choose the left front and right hind hooves?

Amended:

Line 226-228

Since the differences between left and right hoof are meaningless [25], we measured one front hoof and one rear hoof. To take into account the sides of the body, we chose one hoof from the left and one hoof from the right side.